# Monitoring Training Load, Well-Being, Heart Rate Variability, and Competitive Performance of a Functional-Fitness Female Athlete: A Case Study

**DOI:** 10.3390/sports7020035

**Published:** 2019-02-09

**Authors:** Ramires Alsamir Tibana, Nuno Manuel Frade de Sousa, Jonato Prestes, Yuri Feito, Carlos Ernesto, Fabrício Azevedo Voltarelli

**Affiliations:** 1Graduate Program of Health Sciences, Faculty of Medicine, Federal University of Mato Grosso (UFMT), Cuiabá 78000, MT, Brazil; ramirestibana@gmail.com (R.A.T.); favoltarelli@cpd.ufmt.br (F.A.V.); 2Laboratory of Exercise Physiology, Faculty Estacio of Vitoria, Vitoria 29010, ES, Brazil; 3Graduation Program on Physical Education, Catholic University of Brasilia, Brasilia 04534, DF, Brazil; jonatop@gmail.com (J.P.); carlosf@ucb.br (C.E.); 4Department of Exercise Science and Sport Management, Kennesaw State University, Kennesaw, GA 30144, USA; yfeito@kennesaw.edu

**Keywords:** high-intensity functional training, extreme conditioning program, training load, CrossFit

## Abstract

The aim of this case study was to quantify the magnitude of internal load, acute/chronic workload ratio (ACWR), well-being perception, and heart rate variability (HRV) following 38 weeks of functional-fitness training in a female elite athlete. The internal load was obtained with session rating perceived exertion (session-RPE) while the ACWR was calculated by dividing the acute workload by the chronic workload (four-week average). Furthermore, HRV measurements were analyzed via a commercially available smartphone (HRV4training) each morning upon waking whilst in a supine position. The magnitude of internal load was: the weekly mean total during the 38 weeks was 2092 ± 861 arbitrary units (AU); during the preparation for the Open 2018 was 1973 ± 711 AU; during the Open 2018 it was 1686 ± 412 AU; and during the preparation for the Latin America Regional was 3174 ± 595 AU. The mean ACWR was 1.1 ± 0.5 and 50% of the weeks were outside of the ‘safe zone’. The well-being during the 38 weeks of training was 19.4 ± 2.3 points. There were no correlations between training load variables (weekly training load, monotony, ACWR, and HRV), and recuperation subjective variables (well-being, fatigue, sleep, pain, stress, and mood). This case study showed that the training load can be varied in accordance with preparation for a specific competition and ACWR revealed that 50% of the training weeks were outside of the ‘safe zone’, however, no injuries were reported by the athlete. The effectiveness and cost of these methods are very practical during real world functional-fitness.

## 1. Introduction

A relatively new form of exercise referred to as “functional-fitness” (FFT) (also known as high-intensity functional training; extreme conditioning programs) is currently being marketed to a wide range of active (athletes, military) and inactive populations. The competitive FFT (e.g., CrossFit) often consists of a variety of training methods, such as weightlifting/powerlifting, repeated gym bodyweight exercises, cardiovascular exercises, sprints, and flexibility mixed together in order to achieve a high global performance; however, to date, with a lack of monitoring and control of the training methods [1]. The popularity of the FFT training strategy has grown exponentially. Since 2005, the number of CrossFit^®^ affiliates has increased by ~1000-fold (from 1 to >13,000) [2], and competitions have also grown significantly (CrossFit Games, Dubai CrossFit Championship, and Wodapalooza) with prizes higher than $1,000,000 for the winners of several categories (individual and teams). The prize money and the possibility of being sponsored by a sports brand also attracted new athletes to CrossFit^®^, which increased competitiveness. In this way, training periodization and monitoring has become mandatory for achieving great results.

Although FFT has been widely practiced [3], there is currently limited evidence of training load monitoring in athletes engaging in these activities. The application of appropriate training load is one of the fundamental factors for positive physiological adaptations to occur with consequent improvement in performance. Insufficient loads will not result in physiological adaptations, and excessive loads will result in negative adaptations, including non-functional overreaching and/or overtraining [4,5,6].

The most pertinent and well-designed studies about FFT converge and have found that training planning and follow-up are very important to avoid deleterious consequences [1,7,8]. Tibana et al. [9] reported that two consecutive days of training resulted in unfavorable blood cytokine responses (decrease in anti-inflammatory and increase in proinflammatory cytokines) during the 48 hours after exercise, reinforcing the need of sufficient rest for practitioners. Similarly, Heavens et al. [10] found that an adapted protocol from CrossFit (Linda: 10-9-8-7-6-5-4-3-2-1 reps of the triplet deadlift, bench press and clean) increased blood proinflammatory cytokines (after 60 min of exercise) and creatine kinase (after 24 hours of exercise). Drake, Smeed, Carper and Crawford [6] showed that even with an increase of physical fitness after four weeks of FFT, the participants approached a state of functional overreaching. The authors warned that non-functional overreaching could develop if the high intensity was maintained for a long term [6]. Unfortunately, no data was evaluated to explore the relationship between muscle function and training load. 

Considering the wide variety of exercises used in FFT (strength, gymnastics, and endurance), training load quantification is a challenge. Due to the variety of training methods used during FFT, external training load (e.g., speed, pace, distance, repetitions) is a poor tool to monitor the training load. Recently, Tibana et al. [11] and Crawford et al. [12] validated the session rating of perceived exertion (RPE) method during FFT. The session-RPE method was first proposed by Foster [13], and Foster et al. [14] quantified internal training load. This method is one of the main tools to quantify internal training load, detaching low cost and practical application. Moreover, the session-RPE method is sensitive to alterations in training load [5]. Nevertheless, the acute/chronic workload ratio (ACWR; calculated dividing the acute workload by the chronic workload) has been used as a safe and systematic method to progress training loads, and consequently, reduce injury risk [4,15]. The ACWR is a model that provides an index of athlete preparedness. It takes into account the current workload (acute) and the workload that an athlete has been prepared for (chronic). Is this sense, the chronic load is analogous to a state of ‘fitness’ and the acute load is analogous to a state of ‘fatigue’ [4,16]. 

Physiological variables, such as testosterone (T), cortisol (C), T/C ratio, immune biomarkers, creatine kinase, and performance analysis are often described in the literature to control training stress [17]. However, these biochemical measures have a high cost and are not common during daily training practice, albeit the used perceived exertion scales can provide a more practical and cheaper alternative of training load. To complement the perceived exertion scales, questionnaires and diaries are used to determine the effect of training load on behavior responses of athletes (humor states, stress tolerance, well-being, quality of recovery, upper respiratory tract infections, among others) [18]. Lastly, research has supported the use of heart rate variability (HRV) as an objective, physiological indicator of stress and recovery to variations in training load among athletes [19,20,21].

To the best of our knowledge, no previous study quantified training load using the session-RPE method, ACWR, well-being perception, and heart rate variability during a competitive season in an FFT elite athlete. Thus, the aim of the present study was to quantify the magnitude of internal load through session-RPE, ACWR, well-being perception, and HRV following 38 weeks of FFT in an elite female athlete.

## 2. Materials and Methods

### 2.1. Case Report Design

The case report included a female athlete. The athlete was 34 years old, 67 kg of body mass, 155 cm in height, 14% of body fat, and had four years of FFT experience. Her 1-repetition maximum was 130 kg for the back squat, 112 kg for the front squat, 95 kg for the clean and jerk, and 77 kg for the snatch. Prior to the case report, the participant signed a written informed consent, which was approved by the local ethics committee (Protocol number: 2.698.225; 7 June 2018) and fully outlined the purpose, protocols, procedures, and risks associated. Food intake and supplementation were not controlled during the study.

### 2.2. Training Sessions

The athlete completed 188 (5 days/week) training sessions that generally followed the format of: warm-up, traditional multiple-joint, functional, resistance exercises (squat, press, deadlift, Olympic lifts), conditioning, and gymnastic skills (hand stands, ring, bar exercises, etc.). A one-week training log is presented in the Appendix A. The training sessions and the periodization was developed by an experienced FFT coach. During the sessions, the athlete was supported by a coach.

The goal of the conditioning sessions was to complete each training session as fast as possible, without compromising exercise technique. Some exercises were performed for a best time, and others were performed in the As Many Repetitions as Possible (AMRAP) style during a fixed time period.

The season began in September 2017 and ended in May 2018 (nine months) and it was composed of three minor competitions in the 3rd, 7th, and 12th weeks and two major competitions, the CrossFit^®^ Open 2018 between the 25th and 29th weeks, and the CrossFit^®^ Regionals (South America) in the 38th week. Table 1 presents the calendar competitions and the ranking during the season.

### 2.3. Heart Rate Variability

Photoplethysmography was used to acquire HRV measurements via a commercially available smartphone app known as “HRV4training” (see http://www.hrv4training.com) as described elsewhere [22]. The subject was instructed to take one-minute HRV measurement each morning upon waking whilst in a supine position [23]. The weekly mean of log-transformed square root of the mean sum of the squared differences between two R-peak of a traditional ECG heart-beat waveform, the R–R intervals (LnRMSSD), was the HRV measure used for analysis. The average of HRV during the seven days of the week was used as the weekly HRV score.

### 2.4. Quantifying Training Load

Training load, expressed in arbitrary units (AU), was calculated using the session-RPE method proposed by Foster, Florhaug, Franklin, Gottschall, Hrovatin, Parker, Doleshal and Dodge [14], multiplying the total duration of a bout or exercise session in minutes by the training intensity. Intensity was measured by a modified version of Borg’s CR-10 scale of perceived exertion, referred to as session-RPE. The session-RPE score was obtained from the athletes 30 min after the protocol of training. The daily training load was expressed as a single value in arbitrary units (AU). Training load was expressed in weekly training load (the sum of 7 days) and mean training load (average of 7 days). Training monotony was calculated using the ratio between weekly internal training load and its standard deviation [13,14].

### 2.5. Acute/Chronic Workload Ratio (ACWR) Calculation

One-week of training load (session-RPE) data represented an acute workload, while a four-week average of acute workload represented chronic workload (lasting for weeks). The ACWR was calculated by dividing the acute workload by the chronic workload [4,15,24]. A larger acute workload as compared with chronic workload consisted of a high ACWR and a larger chronic workload as compared with the acute workload reflected a low ACWR. 

### 2.6. Well-Being

During the study period, the athlete completed a custom-made psychological questionnaire that was based on the recommendations of McLean et al. [25]. The questionnaire assessed their fatigue, sleep quality, general muscle soreness, stress levels, and mood on a five-point scale (scores of 1 to 5). Overall well-being was then determined by summing the five scores. The well-being questionnaire was completed 24 h after the last training session of the week.

### 2.7. Statistical Analysis

Daily training data of the female athlete was organized into averages or sums corresponding to one week (e.g., weekly training load, the sum of the training load of each day for one week; mean weekly training load, average of training load for one week). Mean values were calculated corresponding to the total training program (38 weeks) or a specific period of training (e.g., during the Open 2018). The Pearson product moment correlation was used to evaluate correlations between training load variables (weekly training load, monotony, ACWR), physiology (LnRMSSD), and recuperation subjective variables (well-being, fatigue, sleep, pain, stress, and mood). SPSS version 20.0 (IBM Corporation, Somers, NY, USA) software was used.

## 3. Results

The weekly training load, the mean training load, and monotony of the training assessed by the session-RPE method are presented in Figure 1. The mean total weekly training load during the 38 weeks was 2092 AU. During the weeks, the lowest training load assessed was 590 AU (week 38; 28% of the mean weekly training load) and the highest training load assessed was 3840 AU (week 36; 184% of the mean weekly training load). The mean monotony was 1.30 (lowest: 0.60; highest: 2.36). It was observed that during the minor and the major competitions and transition phases, the training load was lower than the previous weeks. The mean total weekly training load during the preparation for the Open 2018 was 1973 ± 711 AU, during the Open 2018 was 1686 ± 412 AU, and during the preparation for the Regionals was 3174 ± 595 AU, as shown in Figure 1B. Training monotony during the preparation for the Open 2018 was higher than during the Open 2018. Monotony increased again during the preparation for the Regionals. 

Figure 2 presents the ACWR during the 38 weeks of training. The mean ACWR was 1.1 and values outside of the theoretical ‘safe zone’ (0.8 and 1.3) [4] were observed during the preparation phase for competition. 

The well-being presented an undulatory behavior during the 38 weeks of training (mean value: 19.4) as shown in Figure 3. High values were presented during the minor competitions or the transition phase; however, during Open 2018, the well-being stabilized at approximately 19 points and increased during the transition phase.

The fatigue, sleep, pain, stress, and mood scores, evaluated on a scale of 5 points (1–5 points), and heart rate variability evaluated by the LnRMSSD method are presented in Figure 4. Fatigue and pain scores presented a high variability during the 38 weeks, while sleep, stress, and mood scores had minimal changes during the period. LnRMSSD presented an increase during the preparation for the Open 2018, remained high during the Open 2018, and decreased during the preparation for the Regionals. 

Table 2 presents the mean, standard deviation, minimum, and maximum values for the variables analyzed during the 38 weeks of the study. No correlations were observed between training load variables and subjective variables of well-being or fatigue, sleep, pain, stress, or mood during the 28 weeks of training.

## 4. Discussion

The aim of the present case study was to quantify the magnitude of internal training load through session-RPE, ACWR, well-being perception, and heart rate variability following 38 weeks of FFT in an elite female athlete. Results from the present study revealed that the elite athlete with classification in CrossFit Regionals presented the mean total weekly training load of 2092 AU, and the mean ACWR of 1.1 with 50% of the weeks outside of the theoretical ‘safe zone’. Furthermore, there were no correlations between training load variables (weekly training load, monotony, ACWR), physiology (LnRMSSD), and recuperation subjective variables (well-being, fatigue, sleep, pain, stress, and mood).

Recently, the Consortium for Health and Military Performance (CHAMP) and the American College of Sports Medicine considered FFT as an exercise training modality with an increased risk of injury, and suggested the monitoring of training load to reduce injury risk [26]. However, to our knowledge, only two studies analyzed load distribution in FFT practitioners. Tibana et al. [27] showed in a case study that session-RPE was able to distinguish a different internal training load during tapering, overloading, and recovery following the 11-week training program, and that the mean total week training load was ~1300 AU in two amateur athletes. Moreover, Williams et al. [28] investigated the internal training load (heart rate variability and session-RPE) and the risk of overreaching issues in six CrossFit athletes (two athletes qualified for the CrossFit Regionals) across a 16-week period. The authors showed that the average weekly training loads were 2591 ± 890 AU. Moreover, overuse injury risk increased when ‘low’ LnRMSSD week values were observed, accompanied by a high ACWR, while a high ACWR was well tolerated when the LnRMSSD week was ‘normal’ or ‘high’. In the present case study, internal training load was 2092 AU, similar to the study from Williams, Booton, Watson, Rowland and Altini [28] (2591 ± 890 AU), and larger than the study from Tibana, Sousa and Prestes [27] ~1300 AU, which demonstrates that international competition athletes present higher training load, as expected. In this sense, allowing high chronic training loads (i.e., ‘fitness’), without rapid spikes in workloads (i.e., an ACWR greater than ~1.3) is currently considered the ‘best practice’ approach for optimizing performance, whilst minimizing injury risk in elite sport [4,5]. Williams, Booton, Watson, Rowland and Altini [28] converted ACWR to within-individual z-scores, which makes it difficult to compare with the present study. However, individual athletes’ daily ACWR values were outside (32%) of the previously described ‘safe zone’ (0.8–1.3) during the study period [4]. In the present study, 50% of the weeks were outside of the ‘safe zone’. Although these data may be conflicting, Windt et al. [29] reported that spikes in workloads cause an increase in neuromuscular fatigue, which is associated with elevated injury risk. Physical fitness (muscle strength and aerobic fitness) act as a moderator (i.e., dimmer switch) of the relationship between workload spikes and injury [30,31]. In this case, a spike in workload will elicit a different injury risk dependent on an individual’s fitness level.

Despite the session-RPE method having been able to distinguish different internal training load during the blocks of periodization in these case reports, no correlations were observed between subjective variables nor injury (the athlete had no injuries). In this sense, the present case report was not able to elucidate the relationship between training load and well-being, fatigue, sleep, pain, stress, and mood.

The evaluation of the autonomic control of heart rate via HRV analysis could be an important tool for monitoring individual recovery and training adaptations in elite athletes [32,33]. In the current study there were no correlations between LnRMSSD with training load variables (weekly training load, monotony, ACWR) and recuperation subjective variables (well-being, fatigue, sleep, pain, stress, and mood). This finding is in opposition with the recent work of Flatt et al. [34] who reported that changes in HRV were associated with perceived sleep quality, fatigue, stress, and mood after four weeks among Division-1 sprint-swimmers. Moreover, the authors suggested that the analysis of well-being may have implications for making targeted interventions when decrements in HRV are observed in athletes. A decreased in the LnRMSSD is correlated with lower fitness capacity, higher perceived fatigue, and increased training load. Thus, a correct periodization that attenuates the reduction of the LnRMSSD is very important for the athlete. Considering that both low LnRMSSD and high ACWR were an injury risk marker [28], coaches should be aware of this practical relationship.

With respect to inexpensive, non-invasive, and non-exhaustive measures of assessing fitness and/or wellness (e.g., stress, fatigue), psychological monitoring is also purported to be effective in assessing individual responses to training [35,36]. In this sense, previous studies found that the well-being scale can detect changes in fatigue, muscle soreness, and can be a useful tool in monitoring training-induced stress. McLean, Petrucelli and Coyle [35] observed that when muscle soreness was significantly increased, maximal power output (inertial load cycling technique) was reduced during the next week, suggesting that changes in perceptual fatigue may precede reductions in physical performance in female collegiate soccer players. These results are somewhat in agreement with McGuinness, McMahon, Malone, Kenna, Passmore and Collins [36] who reported substantial changes in players’ well-being to be associated with reductions in running performance in elite female field hockey players. In the present study, there was no correlation between training load variables (weekly training load, monotony, acute ratio) and recovery variables (well-being, fatigue, sleep, pain, stress, and mood). However, different from the previous studies, changes in performance were not measured (e.g., jump height, power output), which may be more associated with changes in well-being than the training load (weekly training load, monotony, acute ratio) per se. Therefore, future studies are required to evaluate changes in well-being and its association with performance.

These findings are of importance in a practical setting, suggesting that coaches could use the well-being scale because it is a very practical and low cost tool for monitoring fatigue during training and competitions. Despite the interesting findings of the present study, some limitations need to be mentioned. First, a case study limits the extrapolation of these findings to athletes/practitioners of FFT. However, Halperin [37] explained that case studies have other advantages that are not commonly recognized: case studies can serve as a potent communication strategy to non-scientist coaches if presented as narratives. Conducting case studies in conjunction with coaches can serve as a “buy-in” strategy which can establish and strengthen relationships between scientists and coaches, which creates possibilities for future research collaborations. Second, we did not assess the change in physical fitness (muscle strength and cardiovascular fitness). On the other hand, we present the results of all competitions of the athlete, which is certainly more significant in the “real world” of these athletes.

## 5. Conclusions

The session-RPE and ACWR methods revealed that 50% of the training weeks were outside of the ‘safe zone’, which reveal an important issue to be controlled by coaches during a training season, and also could be used to prevent injuries and to incorporate changes in training load according to individual responses. Moreover, the use of the well-being scale is recommended, considering that this is a very practical and low cost tool for monitoring fatigue during training and competitions for analyzing the recovery of athletes. The effectiveness and low cost of these methods are very practical during real world functional fitness training.

## Figures and Tables

**Figure 1 sports-07-00035-f001:**
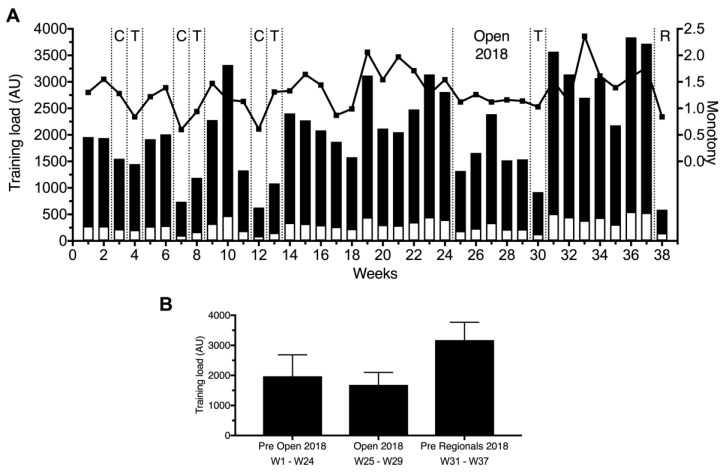
(**A**) Total weekly training load (black bars), mean training load (white bars), and monotony (line) assessed by the session-RPE (rating of perceived exertion) method for 38 weeks. C, minor competitions; T, transition; Open 2018 and R, major competitions. (**B**) Mean weekly training load during preparation of the Open 2018, Open 2018, and preparation of the Regionals 2018.

**Figure 2 sports-07-00035-f002:**
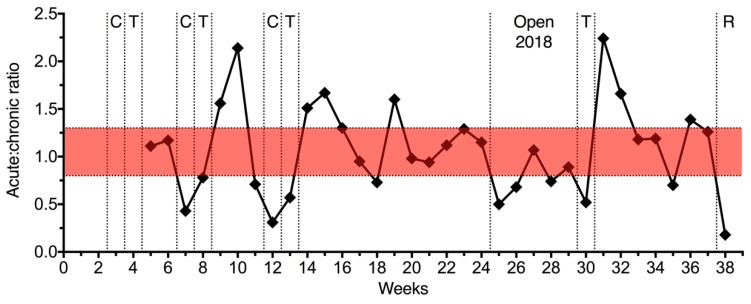
Acute:chronic workload ratio for 38 weeks. C, minor competitions; T, transition; Open 2018 and R, major competitions. The values between 0.8 and 1.3 represent the theoretical ‘safe zone’ [4].

**Figure 3 sports-07-00035-f003:**
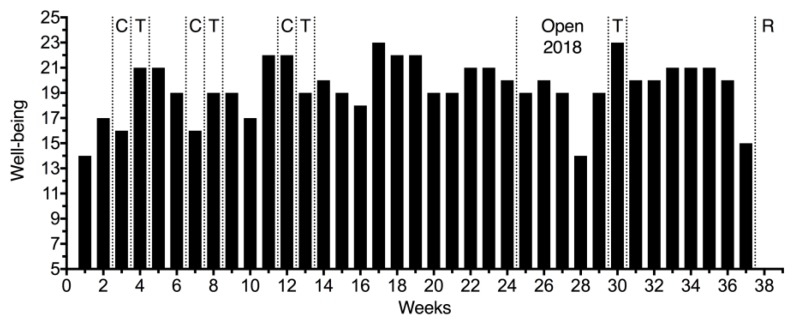
Well-being score (5–25 points) for 38 weeks. C, minor competitions; T, transition; Open 2018 and R, major competitions.

**Figure 4 sports-07-00035-f004:**
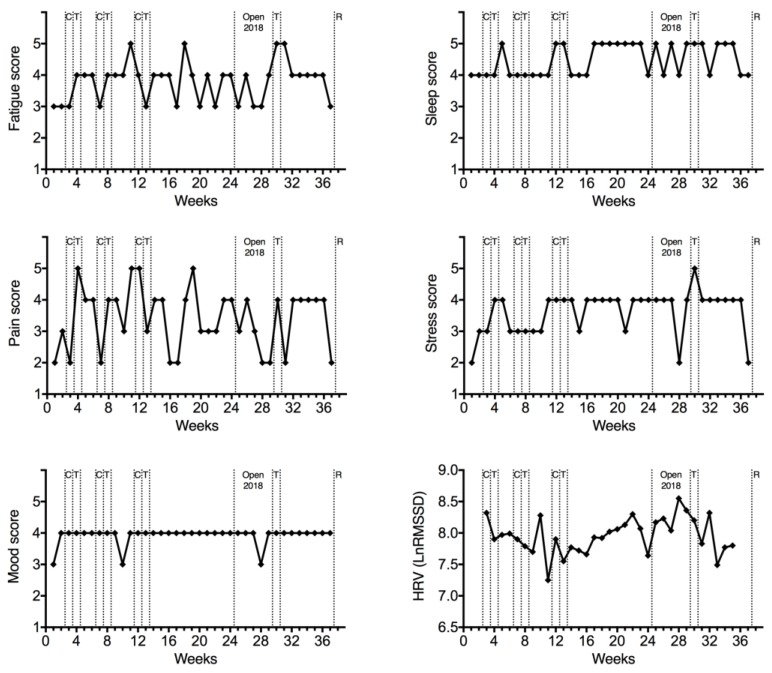
Fatigue, sleep, pain, stress, and mood scores (1–5 points) and heart rate variability (HRV; LnRMSSD) for 38 weeks. C, minor competitions; T, transition; Open 2018 and R, major competitions.

**Table 1 sports-07-00035-t001:** Calendar competitions and ranking during the season.

Month	Competition	Rank
October 2017	Brazil Showdown	3rd
November 2017	Monstar Games	8th
January 2018	WodNation	2nd
February and March 2018	CrossFit Open South America	16th
May 2018	CrossFit Latin America Regional	22nd

**Table 2 sports-07-00035-t002:** Mean, standard deviation (SD), minimum, and maximum values for the training variables during the 38 weeks of the study. AU: arbitrary units; LnRMSSD: log-transformed square root of the mean sum of the squared differences.

	Mean	SD	Minimum	Maximum
Total weekly training load, AU	2092	861	590	3840
Monotony	1.30	0.36	0.60	2.36
Acute:chronic ratio	1.1	0.5	0.2	2.2
Well-being score	19.4	2.3	14.0	23.0
Fatigue score	3.8	0.6	3.0	5.0
Sleep score	4.5	0.5	4.0	5.0
Pain score	3.4	1.0	2.0	5.0
Stress score	3.6	0.7	2.0	5.0
Mood score	3.9	0.3	3.0	4.0
LnRMSSD	8.0	0.3	7.25	8.55

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
