# Peer review of "Monitoring Training Load, Well-Being, Heart Rate Variability, and Competitive Performance of a Functional-Fitness Female Athlete: A Case Study"

_sports, 2019, doi:10.3390/sports7020035_

Round 1

Reviewer 1 Report

This study attempted to document the alterations in training load across a 38 week period in an elite FFT athlete. The study design is appropriate however, the rationale is poorly developed and the discussion makes little attempt to explain the findings. The quality of the written work needs to be improved.

Introduction

L38 unclear what (High-intensity functional training; extreme conditioning programs) relates to

L40 should this be plural i.e. populations? The active and inactive elements would suggest this.

L40 include examples of the ‘others’

L38 to L42 this opening paragraph could do with more background on this training method e.g. what is the aim? How is the training typically programmed?

L43 ‘widely practiced’ can you include a supporting reference. Otherwise this seems anecdotal.

L44 to 47 long sentence. Please consider breaking this down.

L48 it seems you are trying to make the link between training load and fatigue/muscle damage/recovery. Give that blood markers typically display a poor temporal pattern with actual muscle damage can you include any muscle functional data?

L50 below, in the Heavns study, you have detailed that the cytokines are proinflammatory. Is this the same here?

L52 define ‘Linda’

L52 when were these markers taken? i.e. up to 24 hours post

L52 Sentence beginning with ‘Drake’ makes little sense. Also, I think you are trying to draw to much from this study.

L61 unclear what this sentence is trying to say. Do you mean that sRPE is sensitive to alterations in training load?

L63 ACWR needs to be introduced more coherently i.e. how is this method calculated, specifically how can it be used to manage training load

L68 change to ‘albeit the used perceived exertion scales can provide a more practical and cheaper alternative of training load’

L68 ‘thus, questionnaires…’ not sure how this links on to the previous sentence. This sentence also needs a reference.

L74 Unclear what the point of the study is. I get the aim, but what practical application might come out of the data?

L75 you need to introduce HRV and wellbeing perception in the introduction. You also need to explain, explicitly, why external load markers cannot be used. Is the only rationale for why training load needs to be monitored in FFT because of the variety of exercises? Moreover, it would be beneficial to, at the very least, allude a case study approach.

Methods

L80 change to ‘and had four years…’

L80 do you mean ‘her 1-repetition maximum’…?

L85 learning and refining of what? These are detail below but please clarify

L92 please detail the periodization approach. This directly affects the training load.

L97 delete was i.e. the season began in

L98 change composed by

L105 is this a valid and reliable measure of HRV?

L109 unclear what this sentence means

The data analysis section is missing. Even though this is a single subjects design there should be some attempt to analyse the data objectively.

Results

L139 describe how training monotony varied over the 38 weeks. If there was no pattern then explain that

L169 please attempt to describe the data in figure 4. The text in lines L169 and 170 adds nothing.

L177 correlations were performed? Please detail in statistical analysis

Discussion

Rather than make specific points I’ve just highlighted some key issues with the discussion.

L182 to 189 again, the usefulness of the study is unclear. The study has merely documented the training load for 1 athlete.

L190 to 211 there makes no attempt to explain your data but rather provides a rationale for the study

L226 to 234 please explain the fluctuations in HRV in figures 4 and how this relates to the athlete

L247 to 250 this is a major limitations of the current study

L252 how is fatigue defined? What type of fatigue? This application appears rather abruptly

Author Response

The response letter is attached.

Reviewer 2 Report

The Authors emphasize effectiveness and costless of proposed methods. I agree with it. However, according to my opinion, the proposed methods, due to the significant subjectivity of the assessment can be used only in amateur sport.

Comments:

Abstract:

- contains abbreviations which names have not been explained (eg. AU).

Introduction:

-          Line 49 – “Heavens, et al. [8] – it was written incorrectly. It should be “Heavens et al.”.

– line 52-53 –  it was written „Drake, Smeed, Carper and Crawford [4]” – incorrectly. It should be written “Drake et al.” In the further part of the manuscript, such erroneous entries are also made (eg. Line 243).

Materials and Methods:

– age, body mass, height should be written as average values and ranges of variables.

- AU (arbitrary units) – no explanation of how to calculate / estimate/.

References:

– lack of number of pages (eg. 5, 30)

- position 21 – title should also be written in English

Author Response

The response letter is attached.

Reviewer 3 Report

The following study is a case report investigating training load, objective (i.e. HRV) and subjective (i.e. well being questionnaire, RPE) indicators of stress, and ACWR in a competitive FFT elite athlete. Although it is difficult to make recommendations to practitioners based off one individual, the current study holds important implications in athlete monitoring specifically by training load and AWCR. The manuscript is informative and authors are commended on the successful completion of a well-written manuscript. Below are my thoughts that may aid in strengthening the manuscript:

Line 21: change “was” to “were”

Line 31: change “was” to “were”

Line 44: changes “engaged to” to “engaging in”

Line 46: which adaptations are you referring to exactly? Clarify for the reader.

Line 50: remove the “s” from cytokines

Line 53: this sentence needs to be re-worded

Line 63: Need to educate the reader a little more on ACWR

Line 79: Did you gather information about nutritional supplements (if any) were being used? Especially if taken before exercise, many supplements may alter RPE and could also affect your HRV measurements as well.

Line 97: was the season 9 months and the period of time to complete 9 months of competition 21 months? Need further clarification.

Line 135: for week 12, why is there zero mean training load?

Line 152: citation for determining “safe zone” ?

Line 160: this is interesting that AWCR does not reflect well-being of athlete and should be touched on in the dicussion

Line 193: competitions “have” also grown

Line 190-198: I would remove this paragraph and just move this information to the introduction. Referring to the increasing popularity of crossfit is not explaining your results.

Line 226: See below article which will help with the discussion of HRV

Holmes, Clifton, Stefanie Wind, and Michael Esco. "Heart Rate Variability Responses to an Undulating Resistance Training Program in Free-Living Conditions: A Case Study in a Collegiate Athlete." Sports 6.4 (2018): 121.

I think the discussion was very informative

Author Response

The response letter is attached.

Round 2

Reviewer 1 Report

I commend the authors for addressing the comments in a detail manner. Some very typos still present.